# Cd^2+^ Sorption Alterations in Ultisol Soils Triggered by Different Engineered Nanoparticles and Incubation Times

**DOI:** 10.3390/nano13243115

**Published:** 2023-12-11

**Authors:** Karen Manquián-Cerda, Raúl Calderón, Mauricio Molina-Roco, Tamara Maldonado, Nicolás Arancibia-Miranda

**Affiliations:** 1Facultad de Química y Biología, Universidad de Santiago de Chile, Av. B. O’Higgins, 3363, Santiago 9170124, Chile; 2Centro de Investigación en Recursos Naturales y Sustentabilidad, Universidad Bernardo O’Higgins, Fabrica 1990, Segundo Piso, Santiago 8370993, Chile; raul.calderon@ubo.cl; 3Departamento de Acuicultura y Recursos Agroalimentarios, Campus Osorno-Chuyaca, Universidad de los Lagos, Osorno 5290000, Chile; mauricio.molina1@ulagos.cl; 4Instituto de Química, Facultad de Ciencias, Pontificia Universidad Católica de Valparaíso, Av. Universidad 330, Placilla, Valparaíso 2373223, Chile; tamara.maldonado@pucv.cl

**Keywords:** metallic nanoparticles, dose-dependent effects, incubation time, volcanic soils, cadmium

## Abstract

The progressive influx of engineered nanoparticles (ENPs) into the soil matrix catalyses a fundamental transformation in the equilibrium dynamics between the soil and the edaphic solution. This all-encompassing investigation is geared towards unravelling the implications of an array of ENP types, diverse dosages and varying incubation durations on the kinetics governing Cd^2+^ sorption within Ultisol soils. These soils have been subjected to detailed characterizations probing their textural and physicochemical attributes in conjunction with an exhaustive exploration of ENP composition, structure and morphology. To decipher the intricate nuances of kinetics, discrete segments of Ultisol soils were subjected to isolated systems involving ENP dosages of 20 and 500 mg ENPs·kg^−1^ (AgNPs, CuNPs and FeNPs) across intervals of 1, 3 and 6 months. The comprehensive kinetic parameters were unveiled by applying the pseudo-first-order and pseudo-second-order models. At the same time, the underlying sorption mechanisms were studied via the intra-particle diffusion model. This study underscores the substantial impact of this substrate on the kinetic behaviours of contaminants such as Cd, emphasizing the need for its consideration in soil-linked economic activities and regulatory frameworks to optimize resource management.

## 1. Introduction

Soil has become the final destination for multiple pollutants from human activity, which have varied according to the new needs of society [1,2,3]. Depending on their nature, they can affect the balance of different biogeochemical cycles (C, N, P and trace elements, among others), altering the activities that take place in the soil [4,5]. 

The input of different substrates and contaminants for agricultural use in soils is especially relevant, since these soils are vital for obtaining food for human and animal consumption [6,7]. An example of this is the sustained increase in the use of nanotechnology in the agricultural sector, which has meant significant advances, mainly in crop pest control, fertilization and remediation [8,9,10]. For these reasons, in addition to the action of other economic sectors that are not directly linked to agriculture (mainly water treatment, health, and industry), the presence and quantity of engineered nanoparticles (ENPs) in the soil is becoming more abundant every year, causing changes, even as yet unknown ones, in the different dynamics occurring in this matrix [11,12,13].

Examples of this “*nanotechnological irruption*” in this productive sector and other related sectors can be exemplified by the use of three metallic nanoparticles, which are copper, silver and iron nanoparticles (hereafter CuNPs, AgNPs and FeNPs), where the physicochemical, structural and surface characteristics, such as their size, high surface/volume ratio and the presence of highly reactive surface active groups, cause them to interact with the soil components, be it the inorganic fraction, organic matter, or the soil solution, altering the existing equilibrium in the soil [14,15,16].

In the case of CuNPs, their use in agriculture as fungicidal and bactericidal agents is constantly increasing [17,18] due to their use in the prevention and elimination of pests such as *Botrytis cinerea* and *Staphylococcus aureus* [19,20,21]. However, its application is highly inefficient (90% reaches the soil) [22] and, depending on the physicochemical properties of the soil, CuNPs can dissolve or aggregate, causing changes in the dynamics of this metal in soils [23]. Another example to consider is that of AgNPs, whose main entry into the soil is due to external factors. 

The origin of AgNPs in soil is due to their presence in various household products [13,24], which after their useful life inevitably reach soils through irrigation water [25], as well as through the application of sludge from sewage treatment plants, where the retention processes of this type of nanometric substrates are inefficient or non-existent [26]. Due to their more significant toxic effect than CuNPs on different microorganisms, the presence of AgNPs in soils can cause imbalances in the biota of this matrix. These consequences can involve severe environmental risks to agricultural soils [27,28]. Finally, nanoscale zero-valent iron (nZVI) is the most representative example of an FeNP [29,30]. These ENPs have been widely used in the remediation of soils and waters contaminated with organic and inorganic agents [31,32,33]. Their reactivity is strongly conditioned by environmental factors, such as reaction time, O_2_ levels, composition and concentration of ions in the matrix and microbial activity, which favour the passivation or dissolution of nZVI [33]. These phenomena occurring in free nZVI and those interacting with different organic and inorganic soil fractions can cause variability in the disposition of immobilized ions in this new matrix [34].

Depending on the physicochemical properties, such as chemical composition, redox potential, size, morphology, surface area and surface charge, ENPs interact with soil components such as organic matter, clays, Fe oxides and ions that compose the soil solution. In this sense, they participate in, favouring or inhibiting, aggregation, dissolution, sedimentation and element transport processes [35,36,37,38], which can have direct or indirect consequences on soil fertility parameters [22]. Research carried out by Gao et al. determined that parameters such as pH and organic matter (OM) modify the dissolution rate of CuNPs incorporated in Lufa soils but not the soil moisture [23], with the possible consequence that different metals of agricultural importance are displaced by the greater availability of Cu^2+^ [39]. AgNPs have a higher mobility than Ag^+^, which varies depending on the type of OM and mineral fraction of the soil. It has been observed that in clay soils AgNPs are mobilized at depth, but this behaviour changes if the percentage of organic matter is higher, and an upper retention and subsequent dissolution of these materials was observed [13,40]. On the other hand, the high reactivity of nZVI is characterized by their marked redox properties [41,42,43], where, together with functional groups present in OM such as quinones [44,45], they can oxidize or reduce elements present in the soil, altering their availability. In this context, research has shown that these nanoparticles can interact differentially with the fractions that constitute this matrix, causing changes in the existing equilibrium between these components and other ions of agricultural and environmental importance [5,23,46].

Another factor to consider is the agronomic management of soils, which in almost all cases involves the application of fertilizers and, depending on the type of crop and/or soil, can be applied several times throughout the year [47]. For example, volcanic soils, such as Ultisols, which, given their physicochemical characteristics, such as acidic pH, low OM content and a stable inorganic fraction, have a high P retention that is often remedied with the application of phosphate fertilizers, mainly triple super phosphate (TPS). Depending on the origin of the TPS, it can present a variable content of heavy metals, where Cd stands out for its high concentration [48], leading to an increase in the accumulation of heavy metals in the soil [49,50,51]. In particular, Cd is responsible for a number of pathologies affecting humans, such as bone damage, cancer and the itai-itai disease [52]. In addition, it has been determined that this metal, either uncomplexed or complexed, induces the formation of reactive oxygen species (ROS) that produce damage at the root level in crops, affecting their productivity. Consequently, due to the constant applications of phosphate fertilizers, the presence of Cd in soils should be considered from an environmental point of view [53,54]. 

In this context, an eventual higher Cd content and the presence of ENPs in acid soils, such as Ultisols, either by natural or anthropogenic action, is an environmental issue that should be evaluated [32,55,56,57]. This research aims to determine the changes in Cd (the most abundant heavy metal in phosphate fertilizers) sorption in two acidic soils of volcanic origin (Ultisol) for agricultural use. Incubation periods of 1, 3 and 6 months with the presence of different types of nanoparticles and two doses of nanoparticles, 20 mg ENPs·kg^−1^ (low dose) and 500 mg ENPs·kg^−1^ (high dose), were evaluated, where the latter would correspond to a high-risk environmental scenario. The information obtained could help to evaluate the impact of the dose and residence time of ENPs in soils of volcanic origin, and these data could be inputs to consider the development of regulations aimed at the better control of the use of soils and the products that are applied.

## 2. Materials and Methods

### 2.1. Chemical and Nanoparticle Synthesis

The chemical reagents used in the studies were analytical grade AgNO_3_, Cu(NO_3_)_2_·3H_2_O, FeCl_3_·6H_2_O, NaBH_4,_ NaCl, HCl, NaOH and Cd in water 1000 mg·L^−1^ (Titrisol) supplied by Merck (Darmstadt, Germany).

### 2.2. Nanoparticles Synthesis

For the synthesis of the different nanoparticles (CuNPs, AgNPs and FeNPs), procedures described in the literature were used [58,59].

### 2.3. Characterization of ENPs

A characterization of the ENPs was performed, taking into account morphological and surface aspects and using analysis techniques such as scanning electron microscopy (SEM) and electrophoretic mobility (EM). Scanning electron microscopy (SEM) was undertaken using FEI Nova Nano SEM 200 equipment, and particle sizes were observed using the commercial software ImageJ (https://imagej.net/). To recognize the samples, we performed an XRD analysis on a Shimadzu XRD-6000 diffractometer with graphite monochromator and CuKα radiation.

The isoelectric point (IEP) was established by analysing the zeta potential (ZP) via constant stirring of suspensions on a Zeta Meter 4.0 apparatus (Zeta-Meter Inc., Stauton, VA, USA). An amount of 1.0 g of each sample was suspended in 200 mL of a solution with an ionic strength of 1.0 mM (KNO_3_). The IEP was taken from the ZP (mV) vs. pH graph as the pH at which ZP = 0.

### 2.4. Soil Material

Samples were collected from two soils of volcanic origin (Ulitisols) from south–central Chile, located in Collipulli (36°58′ S, 72°09′ W) and Metrenco (38°34′ S, 72°22′ W). The samples were extracted from uncultivated, unfertilized areas to a depth of 15 cm (A horizon). The samples were air-dried, homogenized and sieved (<2 mm) and stored in plastic containers in the dark at 4 °C before physicochemical characterization. The pH was measured and the organic carbon (OC), IEP and cation exchange capacity (CEC) values were obtained for both soils [60]. The total Cd^2+^ content present in the soils was determined by inductively coupled plasma-optical emission spectrometry (ICP-OES, Perkin Elmer Optima 3000) after the samples were dissolved using a microwave digestion process described in the literature [49]. Once the soils were characterized, samples of 500 g of dry soil were incubated, homogenized and maintained in a humid condition at field capacity. The samples were incubated in perforated plastic bags at 20 °C under aerobic conditions in the presence of different types of nanoparticles (CuNPs, AgNPs and FeNPs), considering two doses of 20 and 500 mg ENPs·kg^−1^ throughout periods of 1, 3 and 6 months. To achieve a homogeneous application of the ENPs to the soils, 10 or 250 mg ENPs (according to the dose) were introduced in 100 mL beakers and diluted in distilled water up to approximately 50 mL. They were subsequently sonicated for 20 min to obtain the total dispersion of the ENPs. Finally, the solutions were added to the soil, homogenizing the sample until reaching its field capacity. Once the incubation period was over, sorption tests began.

### 2.5. Sorption Experiments

The Cd^2+^ sorption was investigated through kinetic analysis using 0.01 M KNO_3_ to control the ionic strength of the systems. Samples of 1.00 ± 0.05 g of dry soil, considering the different incubation times (1, 3 and 6 months), were kept in 50 mL centrifuge tubes, and 20 mL of an equilibrating solution of Cd^2+^ de 200 mg·L^−1^ was placed into each tube. The suspensions were then stirred constantly at room temperature (25 ± 2 °C) and kept within the characteristic pH conditions of each soil, which were adjusted with 0.1 M KOH or HNO_3_ using an Orion (model 250 A) pH meter. Once the soil solution suspensions were obtained from the sorption kinetics, they were centrifuged at 2750× *g* for 20 min. The analytical samples were extracted from the suspension at different adsorption times (0–180 min). The supernatants were filtered (0.22 μm hydrophilic PVDF membrane filters) and the concentration of Cd^2+^ in the solution was assessed via ICP-OES (Perkin Elmer Optima 3000). All experiments were undertaken in triplicate.

## 3. Results and Discussion

### 3.1. Nanoparticle Characterization

In general, the morphology observed in the three types of ENPs was spherical (Figure 1), with a high aggregation in the case of FeNPs; a phenomenon frequently reported in the literature as a consequence of the oxidation processes and magnetic properties that characterize these nanoparticles [16]. 

CuNPs and AgNPs showed a lower aggregation due to the absence of magnetic strength [61] and their reduction potentials, which have a lower tendency to form oxides compared with Fe. The average diameters for FeNPs, CuNPs and AgNPs were 46 ± 2, 33 ± 1 and 29 ± 1 nm, respectively (Figure 1). The IEP values showed significant differences (Figure 1d). For example, FeNPs had a positive charge up to pH 7.7 ± 0.2, a value where it was possible to determine its IEP, whereas for CuNPs the IEP value was 3.1 ± 0.3. AgNPs did not show an IEP value throughout the pH range measured (2.5 to 9.5), with a ZP value that ranged from −18.3 to −45.3 mV [61].

### 3.2. Soil Characterization

The main characteristics of these soils are summarized in Table 1. Both soils can be classified as clay soils, where the textural properties determined for Collipulli and Metrenco were 45.6 and 35.3% clay (<0.002 mm), 40.7 and 56.% silt (0.002–0.063 mm), and 13.7 and 8.0% sand (0.063–2.000 mm), respectively [60].

The mineralogical composition of these soils stands out for containing α-cristobalite, goethite, quartz and vermiculite. Kaolinite is the predominant mineral in Collipulli and it is halloysite in Metrenco [60]. The most abundant Fe oxides in these soils are magnetite and goethite, and these soils also have a paramagnetic Fe^3+^ fraction [62]. According to Table 1, both soils have a low concentration of Cd, which classifies them as low-risk soils for the general population and crops, as per current legislation [48,49]. These soils were collected from areas that were neither cultivated nor fertilized, which could explain the low Cd content. The study simulated a scenario of high Cd contamination, which would occur due to extended use of phosphate fertilizers in agricultural management.

### 3.3. Batch Experiment Results

#### 3.3.1. Sorption Kinetics

The dose of 20 mg ENPs·kg^−1^ applied to the soil is based on the permitted amount of Ag^+^ in sludge generated in wastewater treatment plants [24]. In comparison, the dose of 500 mg ENPs·kg^−1^ simulates the worst-case environmental scenario in the range of Ag concentrations from different studies [11,24]. The sorption kinetics of Cd sorption in the control soils and in the soils treated with the different nanoparticles at a dose of 500 mg ENPs·kg^−1^ is shown in Figure 2.

The soils treated with 20 mg ENPs·kg^−1^, displayed a similar behaviour to the control soils, with a slight increase in sorption capacity (*q_e_*) for those soils treated with FeNPs of close to 7%. This behaviour was maintained without significant variations in the different incubation times (Appendix A). The sorption rate (*k*_2_) and sorption capacity (*q_e_*) increased significantly in soils treated with 500 ENPs·kg^−1^ (Table 2) compared with the control soils and those treated with 20 mg ENPs·kg^−1^ (Appendix A). The following decreasing order was observed Soil-FeNPs >> Soil-CuNPs > Soil-AgNPs > Soil for both parameters, which were also sensitive to the incubation process (Figure 2). This suggest that the different ENPs undergo mainly superficial transformations that cause an increase in the amount of reactive sites capable of retaining more Cd^2+^ [63,64]. Similar behaviour has been observed in systems such as those described in this study, where nZVIs significantly increase their As(III) removal capacity with contact time [65]. In the case of AgNPs and CuNPs, contact time has also been found to alter the reactivity of these substrates but to a lesser extent than FeNPs, as reported in different soils [40,66,67]. The Cd sorption percentages for the studied soils, considering the dose of 500 mg ENPs·kg^−1^ were an average of 60 and 85% for CuNPs and FeNPs, respectively. For soils that were treated with AgNPs, only a slight increase of about 10% was observed in comparison with the values reported for the control soils. This result can be attributed to the lower contribution of reactive sites delivered by this nanoparticle as a consequence of its slower oxidation or sulfidation process [63,64]. 

The Cd^2+^ sorption equilibrium in both Ultisols was responsive to the applied treatments, primarily due to variations in the reactivity of the studied nanoparticles. In control soils, the sorption equilibrium was achieved within 200 min for Collipulli soil, whereas it was achieved in less than 30 min for Metrenco. These differences can be attributed to variations in organic matter content, as illustrated in Table 1. Both the introduction of ENPs and the duration of soil incubation led to a reduction in these equilibrium times. This effect was especially pronounced in the Collipulli treatments, where soils treated with FeNPs exhibited the most significant sensitivity to these factors. One plausible explanation for these behaviours is that ENPs, especially FeNPs, may accumulate in areas of higher accessibility, facilitating the faster sorption of Cd^2+^, as shown in Figure 2.

#### 3.3.2. Kinetic Modelling: Pseudo-First-Order (PFO) and Pseudo-Second-Order (PSO) Models

To describe the sorption kinetics of the experimental data, the PFO and PSO models were applied (Figure 3 and Figure 4). Based on the values of the correlation coefficient (*r*^2^) and reduced chi-square statistic (*χ*^2^), the PSO model showed a better fit to the experimental data (Table 2), suggesting that, on the surface of the soil and the soil treated with the ENPs, Cd is preferentially adsorbed by a chemisorption [68,69]. The better fit of the PSO model could be explained by the boundary conditions that describe this model and were discussed in detail by Azizian 2004 [70]. From the theoretical point of view, in the mathematical derivation of both models the term *θ* appears, which represents the difference between the initial concentration and the equilibrium concentration (C_0_–C_e_), and *θ* is the value of the equilibrium coverage fraction. Both terms depend strongly on the experimental conditions [69].

It has been determined that when the initial solute concentration is high, the term θ can be ignored from the general kinetics expression, resulting in the expression of the PFO model [70,71]. However, with the incorporation of ENPs into soils, an increase in Cd retention was observed that was attributed to the presence of new sorption sites from the nanoparticles in addition to those already existing in the soil. 

This phenomenon indicates that the concentration of Cd^2+^ to be lower than the number of sites available in these treatments (soil-ENPs) with respect to the control soils. So, the term *βθ* cannot be ignored, a condition described in the PSO model [70]. This phenomenon is readily evident in soils treated with FeNPs, as illustrated in Figure 4. FeNPs demonstrate heightened reactivity and increased vulnerability to environmental conditions. This behaviour has been well documented in soils contaminated with As, Cd, Pb and Zn, where the presence of Fe nanoparticles enhances the stability of these pollutants [16,32,72]. The pseudo-second-order model also allows for the calculation of the initial adsorption rate, as *qt/t* approaches 0 is defined as *h* = *k·q_e_^2^*. The values of this parameter for Metrenco soil treated with FeNPs were 10 times higher than the control soils (Table 2, Figure 4). This could be dependent on the higher percentage OM that these soils possess, making the nanoparticles more prone to surface transformations, such as the formation of oxides, oxyhydroxides or sulphides, which increased with longer incubation times [59,73]. 

#### 3.3.3. Solute Transport Mechanism: Intraparticle Diffusion Kinetic Model

The intraparticle diffusion model was applied to the experimental data in order to interpret the different Cd^2+^ sorption processes from a mechanistic point of view in the studied soils and to analyse how the presence of different ENPs and incubation times altered Cd^2+^ removal in both Ultisols [68,74] (Table 3 and Appendix A). The initial segment signifies surface diffusion (Step I), followed by the intraparticle diffusion mechanism denoted by the second linear phase (Step II). The third stage (Step III) elucidates the migration from macropores to micropores within the systems under examination [69]. Cd^2+^ sorption in all treatments occurs by the three processes described by the Weber–Morris model (Figure 5), where the first sorption process of this model represents surface or film diffusion [75,76].

This sorption mechanism is the one that predominates in all the treatments studied and is more significant in those soils treated with FeNPs and CuNPs, suggesting that these nanoparticles could be retained in easily accessible areas and cause Cd^2+^ to be retained in the soil and nanoparticle sites [76]. The second section of these graphs (Figure 5) describes the intraparticle diffusion, which is the limiting step. In this stage, no major variations were observed, indicating that the type of nanoparticles or the incubation time do not cause a major impact on this process. The intraparticle diffusion model allows for the calculation of the thickness of the limiting layer (*C*) associated with instantaneous adsorption [75,77]. It was observed that the aging time of ENPs alters the *C*_2_ values, where it was determined that this parameter possesses the following decreasing sequence Soil-FeNPs >> Soil-CuNPs ≥ Soil-AgNPs ≥ Soil, where the larger intersections suggest that surface diffusion plays an important role as a rate-limiting step for Cd in Soil-FeNPs [75,77].

### 3.4. Cadmium Sorption: Role of Metallic Nanoparticles

The results obtained in this study allowed us to determine the role of ENPs in the sorption of Cd^2+^ in volcanic soils with acidic characteristics, where the reactivity of these substrates generates changes in the chemical environments of the soil, causing the sorption of this heavy metal to be greater in those ENPs that are more reactive in these types of soils [55]. The surface factors, such as the nature of the ENPs’ active sites, chemical hardness and surface charge, have a determining influence on their capacity to interact with Cd^2+^ and the soil components [11,28,75]. For example, Fe nanoparticles, given their redox character, can alter groups that are part of the soil organic matter, such as quinones, carboxyl acids and phenols, mainly causing this soil fraction to have a higher sorption capacity [37,38,44]. However, the sorption of Cd^2+^ and probably of other heavy metals decreases when the soil interacts with less reactive ENPs that have less or no redox properties, such as Cu and Ag, since these substrates only provide functional groups as a sorption mechanism and their interactions with Cd^2+^ are weaker [5,11,17].

## 4. Conclusions

The type of engineered nanoparticles (ENPs) proved to be the most crucial factor influencing the sorption of Cd^2+^ in soils of volcanic origin, surpassing the impact of ENP dose and incubation time. The FeNPs resulted in faster and higher Cd retention in all studied soils and treatments, which was attributed to the high reactivity of these substrates. The soil conditions further facilitated surface transformations of the FeNPs. However, kinetic equations describing solute transport indicated that Cd removal occurred in more energetic sites in the Soil-AgNPs and Soil-CuNPs treatments.

The results provide a preliminary understanding of how these nanoparticles modify the sorption kinetics of heavy metals, specifically Cd. The differences in behaviour could be explained by the varying chemical environments in which the nanoparticles are submerged in soils and influenced by factors such as the type and content of organic matter and the characteristics of the inorganic fraction. These factors trigger distinct responses in the reactivity of the ENPs associated with redox reactions.

These data and their analyses offer valuable insights for managing contaminated soil or soil with potential loads of ENPs and heavy metals. This information enables the projection of an appropriate strategy for agricultural use.

## Figures and Tables

**Figure 1 nanomaterials-13-03115-f001:**
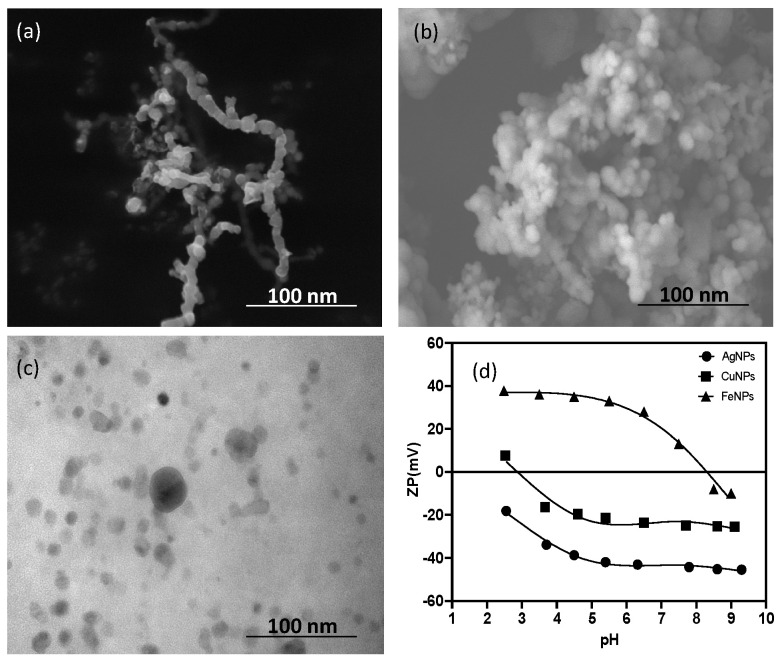
SEM images (**a**) FeNPs, (**b**) CuNPs (**c**) AgNPs and (**d**) zeta potential (mV) versus the pH curves of ENPs.

**Figure 2 nanomaterials-13-03115-f002:**
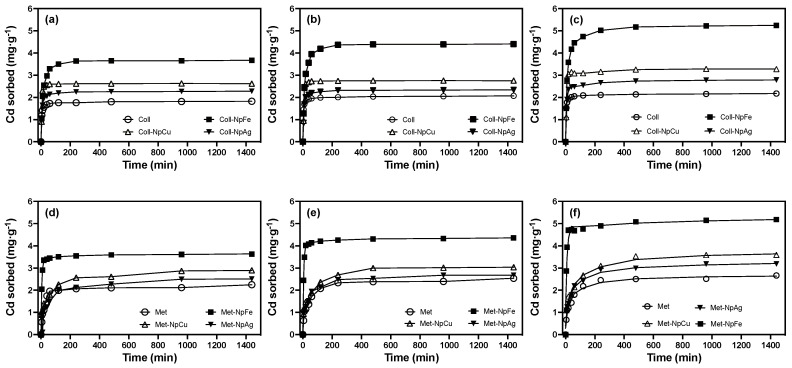
Effect of nanoparticle type and aging time on Cd sorption kinetics in Ultisol soils at a dose of 500 mg ENPs·kg^−1^. (**a**,**d**) Collipulli and Metrenco at 1 month, (**b**,**e**) Collipulli and Metrenco at 3 months and (**c**,**f**) Collipulli and Metrenco at 6 months.

**Figure 3 nanomaterials-13-03115-f003:**
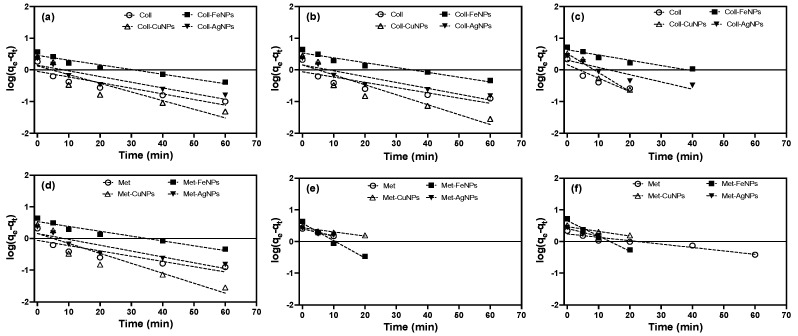
PFO kinetics for the sorption of Cd in Ultisol soils with a dose of 500 mg ENPs·kg^−1^. (**a**,**d**) Collipulli and Metrenco at1 month, (**b**,**e**) Collipulli and Metrenco at 3 months and (**c**,**f**) Collipulli and Metrenco at 6 months.

**Figure 4 nanomaterials-13-03115-f004:**
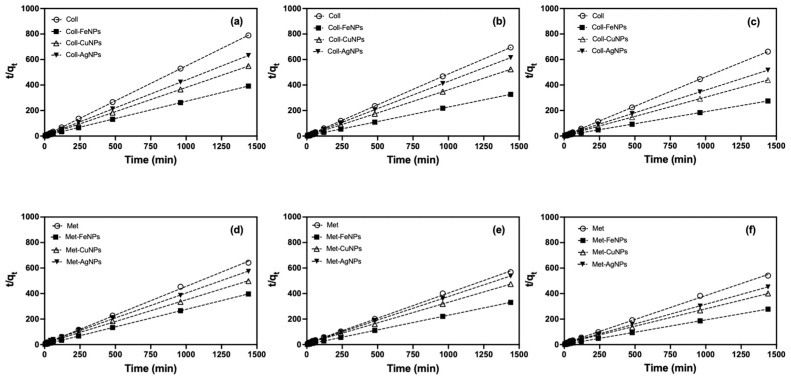
PSO kinetics for the sorption of Cd in Ultisol soils with a dose of 500 mg ENPs·kg^−1^. (**a**,**d**) Collipulli and Metrenco at 1 month, (**b**,**e**) Collipulli and Metrenco at 3 months and (**c**,**f**) Collipulli and Metrenco at 6 months.

**Figure 5 nanomaterials-13-03115-f005:**
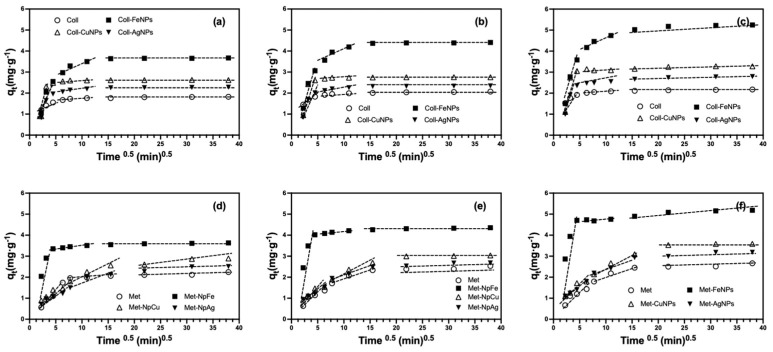
Fits of the experimental data to the Weber–Morris kinetic model for the adsorption of Cd in Ultisol soils with a dose of de 500 mg ENPs·kg^−1^. (**a**,**d**) Collipulli and Metrenco at 1 month, (**b**,**e**) Collipulli and Metrenco at 3 months and (**c**,**f**) Collipulli and Metrenco at 6 months.

**Table 1 nanomaterials-13-03115-t001:** Main parameters of the Ultisol soils. Analyses were carried out in triplicate and the standard error was less than 3%.

Parameter	Collipilli	Metrenco
Soil Order or Source	Ultisol	Ultisol
Soil Class	Fine, mixed,	Fine,
	thermic typic rhodoxeralf	mesic paleumult
Sampling Location		
Latitude	38°58′ S	38°34′ S
Longitude	72°09′ W	72°22′ W
Rainfall (m·year^−1^)	120–400	100–300
Mean annual temperature (°C)	15.8	14.6
Electrical conductivity (dSm^−1^)	0.040 ± 0.001	0.050 ± 0.001
Organic carbon (wt%)	2.1 ± 0.2	2.8 ± 0.1
Available P (mg·kg^−1^)	4.0 ± 0.1	10 ± 0.2
Total P (mg·kg^−1^)	821 ± 8	807 ±15
Available Cd (mg·kg^−1^)	<0.01	<0.01
Total Cd (mg·kg^−1^)	0.030 ± 0.01	0.027 ± 0.01
pH (H_2_O)	6.2 ± 0.1	5.8 ± 0.1
Exchangeable cations (cmol(+)·kg^−1^)	6.0 ± 0.1	8.0 ± 0.1
Calcium	7.0 ± 0.1	10.8 ± 0.1
Magnesium	3.6 ± 0.1	3.2 ± 0.1
Potassium	0.55 ± 0.0	1.1 ± 0.0
Sodium	0.1 ± 0.0	0.2 ± 0.0
IEP	1.7 ± 0.2	0.18 ± 0.1
Mineralogical composition > 50%	Kaolinite	Halloysite

**Table 2 nanomaterials-13-03115-t002:** Kinetic parameters predicted from the pseudo-second-order model for control soils and soils treated with 500 mg ENPs-kg^−1^.

Collipulli												
	1 Month				3 Month				6 Month			
Treatment	Control	AgNPs	CuNPs	FeNPs	Control	AgNPs	CuNPs	FeNPs	Control	AgNPs	CuNPs	FeNPs
*q_exp_ * (mg·g^−1^)	1.7 ± 0.1	2.2 ± 0.1	2.6 ± 0.1	3.5 ± 0.3	2.0 ± 0.1	2.3 ± 0.1	2.7 ± 0.3	3.9 ± 0.3	2.1 ± 0.3	2.6 ± 0.4	3.1 ± 0.3	3.9 ± 0.1
*q_exp_ * (%)	44.2 ± 2.4	54.9 ± 3.2	65.1 ± 4.1	87.4 ± 1.3	49.8 ± 3.5	56.4 ± 4.6	68.4 ± 4.3	97.0 ± 2.6	52.2 ± 2.5	63.7 ± 3.5	77.3 ± 4.6	95.0 ± 3.7
Parameters	1 Month				3 Month				6 Month			
*q_e_ * (mg·g^−1^)	1.8 ± 0.1	2.3 ± 0.1	2.7 ± 0.1	3.3 ± 0.1	2.0 ± 0.0	2.4 ± 0.1	2.9 ± 0.1	4.2 ± 0.1	2.1 ± 0.1	2.8 ± 0.1	3.3 ± 0.1	4.0 ± 0.1
*k*_2_ (×10^−4^ g·mg^−1^·min^−1^)	0.2 ± 0.1	0.1 ± 0.0	0.1 ± 0.0	0.1 ± 0.0	0.2 ± 0.0	0.1± 0.0	0.1 ± 0.0	0.0 ± 0.0	0.2 ± 0.0	0.1 ± 0.0	0.1 ± 0.0	0.0 ± 0.0
*h* (mg·g^−1^·min^−1^)	0.7 ± 0.0	0.4 ± 0.0	0.6 ± 0.0	0.4 ± 0.0	1.0 ± 0.1	0.4 ± 0.0	0.7 ± 0.0	0.5 ± 0.0	1.0 ± 0.0	0.5 ± 0.0	0.7 ± 0.0	0.5 ± 0.0
*r* ^2^	0.999	0.975	0.917	0.994	0.999	0.977	0.925	0.996	0.983	0.942	0.948	0.991
*χ* ^2^	0.019	0.115	0.072	0.013	0.015	0.098	0.087	0.012	0.01	0.071	0.479	0.115
Metrenco												
Treatment	Control	AgNPs	CuNPs	FeNPs	Control	AgNPs	CuNPs	FeNPs	Control	AgNPs	CuNPs	FeNPs
*q_exp_ * (mg·g^−1^)	2.1 ± 0.2	2.3 ± 0.4	2.6 ± 0.4	3.6 ± 0.3	2.3 ± 0.4	2.5 ± 0.3	2.7 ± 0.2	4.1 ± 0.1	2.5 ± 0.3	3.0 ± 0.4	3.5 ± 0.3	4.1 ± 0.5
*q_exp_ * (%)	57.3 ± 2.4	56.8 ± 1.9	65.3 ± 4.3	89.8 ± 7.5	58.4 ± 5.1	61.9 ± 3.9	67.9 ± 4.4	96.5 ± 4.9	62.5 ± 4.2	74.7 ± 4.9	88.3 ± 7.7	97.8 ± 2.1
Parameters	1 Month				3 Month				6 Month			
*q_e_ * (mg·g^−1^)	2.2 ± 0.1	2.5 ± 0.1	2.8 ± 0.1	3.7 ± 0.1	2.4 ± 0.1	2.5 ± 0.1	2.8 ± 0.3	4.2 ± 0.1	2.6 ± 0.1	3.1 ± 0.1	3.5 ± 0.2	4.3 ± 0.1
*k*_2_ (×10^−4^ g·mg^−1^·min^−1^)	0.1 ± 0.0	0.1 ± 0.0	0.1 ± 0.0	0.1 ± 0.0	0.02 ± 0.00	0.02 ± 0.00	0.01 ± 0.00	0.1 ± 0.0	0.02 ± 0.00	0.02 ± 0.00	0.01 ± 0.00	0.1 ± 0.0
*h* (mg·g^−1^·min^−1^)	0.2 ± 0.0	0.1 ± 0.0	0.1 ± 0.0	1.2 ± 0.0	0.1 ± 0.0	0.1 ± 0.0	0.1 ± 0.0	1.4 ± 0.0	0.1 ± 0.0	0.2 ± 0.0	0.1 ± 0.0	1.5 ± 0.0
*r* ^2^	0.983	0.971	0.948	0.972	0.97	0.982	0.968	0.975	0.971	0.988	0.941	0.949
*χ* ^2^	0.095	0.048	0.045	0.012	0.021	0.052	0.035	0.017	0.023	0.083	0.054	0.029

**Table 3 nanomaterials-13-03115-t003:** Kinetic parameters forecasted from the linear evaluation of the intraparticle diffusion kinetic model for the soils studied.

Treatment	Control	AgNPs	CuNPs	FeNPs	Control	AgNPs	CuNPs	FeNPs	Control	AgNPs	CuNPs	FeNPs
Parameters	1 Month				3 Month				6 Month			
*q_e_*_-2_ (mg·g^−1^)	1.6 ± 0.3	2.0 ± 0.2	2.4 ± 0.1	2.6 ± 0.4	1.8 ± 0.2	2.0 ± 0.3	2.6 ± 0.1	3.1 ± 0.2	1.9 ± 0.2	2.4 ± 0.1	3.1 ± 0.1	3.6 ± 0.1
*k_int-_*_2_ (mg·g^−1^·min^1/2^)	0.2 ± 0.0	0.5 ± 0.0	0.7 ± 0.0	0.7 ± 0.0	0.2 ± 0.0	0.5 ± 0.0	0.7 ± 0.0	0.7 ± 0.0	0.2 ± 0.0	0.6 ± 0.0	0.8 ± 0.0	0.9 ± 0.0
*C*_2_ (mg·g^−1^)	0.9 ± 0.0	0.1 ± 0.0	0.2 ± 0.0	0.1 ± 0.0	1.1 ± 0.0	0.0 ± 0.0	0.1 ± 0.0	0.0 ± 0.0	1.2 ± 0.1	0.0 ± 0.0	0.0 ± 0.0	0.1 ± 0.0
*r* ^2^	0.964	0.859	0.819	0.922	0.941	0.883	0.799	0.922	0.94	0.859	0.915	0.948
Metrenco												
Parameters	1 Month				3 Month				6 Month			
*q_e-_*_2_ (mg·g^−1^)	1.1 ± 0.1	1.1 ± 0.1	1.4 ± 0.0	3.4 ± 0.3	1.1 ± 0.0	1.2 ± 0.1	1.5 ± 0.0	3.7 ± 0.1	1.2 ± 0.1	1.4 ± 0.0	1.7 ± 0.0	4.0 ± 0.0
*k_int_*_-2_ (mg·g^−1^·min^1/2^)	0.2 ± 0.0	0.1 ± 0.0	0.2 ± 0.0	0.6 ± 0.0	0.2 ± 0.0	0.2 ± 0.0	0.3 ± 0.0	0.6 ± 0.0	0.2 ± 0.0	0.1 ± 0.0	0.3 ± 0.0	0.8 ± 0.0
*C*_2_ (mg·g^−1^)	0.1 ± 0.0	0.4 ± 0.0	0.3 ± 0.0	0.9 ± 0.1	0.2 ± 0.0	0.7 ± 0.0	0.4 ± 0.0	1.0 ± 0.1	0.2 ± 0.0	0.8 ± 0.0	0.4 ± 0.0	1.2 ± 0.0
*r* ^2^	0.905	0.89	0.994	0.922	0.81	0.953	0.994	0.922	0.811	0.993	0.975	0.996

## Data Availability

Data are contained within this article.

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
