# Peer review of "Cd2+ Sorption Alterations in Ultisol Soils Triggered by Different Engineered Nanoparticles and Incubation Times"

_nanomaterials, 2023, doi:10.3390/nano13243115_

Round 1
Reviewer 1 Report
Comments and Suggestions for Authors
Journal: Nanomaterials
Title: Cd2+ Sorption Alterations in Ultisol Soils Triggered by Different Engineered Nanoparticles and Incubation Time
MS No. nanomaterials-2727977
The investigation is geared towards unravelling the implications of an array of ENP types, diverse dosages, and varying incubation durations on the kinetics governing Cd2+ sorption within Ultisol soils. The comprehensive kinetic parameters were unveiled by applying the pseudo-first order and pseudo-second order models. At the same time, the underlying sorption mechanisms were studied via the intra-particle diffusion model.
The topic is interesting due to the wide application of NPs and its effects on soil environment. However, the basis on which the experiments were designed is not so clear. The mechanisms on the effect of NPs on adsorption of Cd are also not clear. The paper should be revised largely.
1. As shown in Table 1, the intrinsic concentrations of Cd in soils (0.030 and 0.027 mg/kg) are not over the standard limited values (in the most countries). Thus, it seems that the investigation on the adsorption alteration of Cd is not significant.
2. L205, 500 mg ENPs/kg, is the designed value too high, according the practice? Especially Cu and Ag, because they are heavy metal contaminants to soil.
3. L205 and 236, format of “Fig.” or “Figure”;
4. In tables and main text, italic format for variation quantity symbols;
5. L240 and 302, the series numbers of table are confused;
6. Table 3, explain why the values of qexp (%) are larger than 100%;
7. The mechanisms of effect on adsorption of Cd are not clear. The readers do not know the surface composition of NPs, the interaction between NPs and soil particles, and so on.
Author Response
Comment: As shown in Table 1, the intrinsic concentrations of Cd in soils (0.030 and 0.027 mg/kg) are not over the standard limited values (in the most countries). Thus, it seems that the investigation on the adsorption alteration of Cd is not significant.
Answer: Understanding the reviewer's comment, it is evident that the Cd content in these soils is low, as the soils used in this study are non-agricultural soils, which is why they were selected to see the effect of high doses of Cd, which would be associated with applications of phosphate fertilisers, as described by several investigations, highlighting the legacy of this type of fertiliser. Modifications were made to the text in order to improve the information provided.
Comment: L205, 500 mg ENPs/kg, is the designed value too high, according the practice? Especially Cu and Ag, because they are heavy metal contaminants to soil.
Answer: The ENPs selected, mainly those of Ag and Cu, were taken from the literature, which considered the amounts of both metals allowed in sludge generated in wastewater treatment plants (20 mg ENPs/kg). However, the amounts of 500 mg ENPs/kg were selected because the soils under study have a marked sorption capacity for different analytes, so the amounts were increased to see a behaviour change. Some of the text was modified for clarity.
Comment: L205 and 236, format of “Fig.” or “Figure” and In tables and main text, italic format for variation quantity symbols.
Answer: Appreciate the reviewer's comment; both words were changed and the tables were modified respecting the journal's format.
Comment: L240 and 302, the serial numbers of the tables are confused
Answer: The tables were edited to avoid interpretation errors. The reviewer's comment is gratefully acknowledged.
Comment: Table 3, explain why the values of qexp (%) are larger than 100%.
Answer: We regret that error; the data were corrected, and we thank the reviewer for alerting us to that situation in the new version of the data summarised in the tables, which were revised and corrected.
Comment: The mechanisms of effect on adsorption of Cd are not clear. The readers do not know the surface composition of NPs, the interaction between NPs and soil particles, and so on.
Answer: The reviewer's observation is true, so we have modified the text in the relevant sections so that readers can understand the information obtained from our research.
Reviewer 2 Report
Comments and Suggestions for Authors
Dear authors,
thank you very much for the well conducted research!
Was there any competition between Cd and phosphat ions in adsorption? I am saying that since we have observed in our studies in remmediation of highly contaminated soils with Fe powder in Georgia (soil pH > 7) a strong P - and B adsorption which induced B deficiency in plants. May be that you should mentioned that in your conclusion.
Otherwise in line 77: What is a Lufa soil?
Letters in Fig 5 are missing!!
Regards
Author Response
Comment: Was there any competition between Cd and phosphat ions in adsorption? I am saying that since we have observed in our studies in remmediation of highly contaminated soils with Fe powder in Georgia (soil pH > 7) a strong P - and B adsorption which induced B deficiency in plants. May be that you should mentioned that in your conclusion.
Answer: We appreciate the reviewer's comments. In other research we are working on, we have noticed that the availability of phosphate, measured as Olsen-P (available phosphorus), fluctuates due to the presence of Cu nanoparticles, something we are studying in greater detail, since P in acid soils such as ours is strongly bound to the inorganic fraction and organic matter, causing its availability to decrease, so the presence of NPs generates temporary changes in the availability of this nutrient, which has consequences on plant development.
In this situation, Fe, in practically all its forms, has a high affinity for the different oxyanions, such as As(V), As(III), Se(VI), Se(IV) and B(III), although not so strongly for the latter. Given the above, if the phosphate concentration is low or pH conditions are favourable for P to be mostly available, the B deficiency could be due to the sorption of this element in the Fe powders. It would be good to consider the pH factor, which is a high value for P, which lowers its interaction with the Fe powder, making it easier for B to retain by this substrate.
Comment: Otherwise in line 77: What is a Lufa soil?
Answer: Standard soils are mainly used for GLP (Good Laboratory Practice) compliant studies, exploring leaching, degradation, metabolism, impact on soil microflora and fauna and adsorption/desorption characteristics of pesticides. These soils are recommended in the German JKI guidelines and other relevant standards (OECD) based on specific characteristics. In addition, our standard soils are used in various non-pesticide-related experiments.
Comment: Letters in Fig 5 are missing!!
Answer: We thank the reviewer for his comments. All figures were revised and corrected, with particular emphasis on Fig. 5.
Round 2
Reviewer 1 Report
Comments and Suggestions for Authors
The reviwer thinks the authors have responded the comments and revised the paper well at large.
The paper could be accepted.